# Cut-Off Value of Voluntary Peak Cough Flow in Patients with Parkinson’s Disease and Its Association with Severe Dysphagia: A Retrospective Pilot Study

**DOI:** 10.3390/medicina59050921

**Published:** 2023-05-11

**Authors:** Kyeong-Woo Lee, Sang-Beom Kim, Jong-Hwa Lee, Seong-Woo Kim

**Affiliations:** Department of Physical Medicine and Rehabilitation, College of Medicine, Dong-A University, Busan 49201, Republic of Korea; kwlee65@hanmail.net (K.-W.L.); sbkim@dau.ac.kr (S.-B.K.); jhlee08@dau.ac.kr (J.-H.L.)

**Keywords:** Parkinson’s disease, aspiration, dysphagia, peak cough flow

## Abstract

*Background and Objectives.* Swallowing and coughing reflexes are both closely associated with airway protection. Peak cough flow (PCF) is associated with dysphagia in several neurogenic diseases. In this study, we aimed to analyze the relationship between PCF and aspiration in Parkinson’s disease (PD) and determine the cut-off value of PCF. *Materials and Methods*. We retrospectively analyzed the records of patients with PD who underwent a videofluoroscopic swallowing study and checked for PCF. A total of 219 patients were divided into an aspiration group (*n* = 125) and a non-aspiration group (*n* = 94). *Results*. Significantly lower PCF values were observed in the aspiration group compared to the non-aspiration group (132.63 ± 83.62 vs. 181.38 ± 103.92 L/min, *p* < 0.001). Receiver operating characteristic curve analysis revealed that a PCF cut-off value of 153 L/min (area under the receiver operating characteristic curve, 0.648; sensitivity, 73.06%; specificity, 51.06%) was associated with aspiration in PD. Additionally, a univariate analysis showed that the male sex, lower body mass indexes, higher Hoehn and Yahr scales, and PCF values of ≤153 L/min indicated an increased risk of aspiration. *Conclusions*. Through a multivariate analysis, we demonstrated that a PCF value ≤153 L/min was associated with an increased risk of aspiration (odds ratio 3.648; 1.797–7.407), highlighting that a low PCF is a risk factor for aspiration in patients with PD.

## 1. Introduction

Parkinson’s disease (PD) is one of the most common neurological disorders worldwide. With disease progression, patients with PD exhibit respiratory muscle weakness and cough impairment due to degenerative damage to the nervous system [1,2,3]. Previous studies have shown that coughing and swallowing are closely linked through neurological mechanisms that operate in a highly coordinated sequence [4,5]. Coughing and swallowing are controlled by central pattern generators (CPGs) of respiration and swallowing in the brainstem [6]. A shared network of brainstem respiratory motor neurons coordinates coughing and swallowing, thereby facilitating airway protection [5]. Degeneration of the ongoing neural pathways, such as in the case of patients with PD, eventually reduces the physiological functions of airway protection, including breathing, coughing, and swallowing [5,7,8].

Respiratory dysfunction in patients with PD may be associated with accessory respiratory muscles, uncoordinated breathing control, and increased chest wall stiffness [1]. Moreover, respiratory muscle weakness has been reported to cause deterioration of pulmonary function, limiting the inhaled volume and intrathoracic pressure for coughing [9,10]. A cough reflex is the most important mechanism of airway protection to discharge foreign substances from the lower airway [10,11], and coughing is a complex physiological reflex that involves multiorgan coordination via complex neural circuits [10,11]. In the absence of the cough reflex, aspiration into the lower airways and lungs cannot be prevented, which may lead to serious complications such as pneumonia [10,12,13,14,15]. Furthermore, respiratory dysfunction in patients with PD leads to inhibition of the cough reflex with disease progression.

Functional associations between coughing and swallowing further complicate the clinical presentation [14]. Dysphagia, which usually occurs in the mid to late stages of PD, affects approximately 80% of patients depending on its course [16]. It can cause nutritional imbalances or respiratory complications, significantly affecting morbidity and mortality [12,15]. Similar to cough disorders, dysphagia in PD is characterized by oropharyngeal bradykinesia and rigidity, incomplete cricopharyngeal relaxation, reduced cricopharyngeal opening, and delayed onset of the swallowing reflex [17,18,19,20]. Additionally, the coordination between swallowing and respiration that enables safe swallowing is impaired in patients with PD [21,22].

Several studies have attempted to predict aspiration by investigating respiratory function [23]. For instance, Pitts et al. [24] analyzed voluntary cough variables in patients with PD and suggested that objective airflow measures from voluntary coughing may predict aspiration. Through voluntary peak cough flow (PCF), it is possible to predict whether the respiratory muscles are involved indirectly. PCF is widely used as a simple measurement method that can sensitively predict changes in the respiratory muscles [14,15]. Silverman et al. [14] reported that in patients with PD, the PCF rate was correlated with the severity of the disease and the penetration–aspiration scale (PAS) score obtained from the video fluoroscopic swallowing study (VFSS) [14]. Furthermore, voluntary PCF reflects airway integrity and disease severity in PD [14].

Previous studies have correlated PCF with the severity of dysphagia in PD [14]; however, no study has identified the cut-off value of PCF for aspiration in VFSS. In our retrospective pilot study, we hypothesized that PD patients with low PCF will have a higher risk of aspiration and determine the cut-off value of PCF for aspiration.

## 2. Materials and Methods

### 2.1. Participants

We retrospectively analyzed the medical records of the patients with PD who underwent VFSS at our tertiary hospital rehabilitation clinical center between December 2017 and December 2020. Patients were diagnosed with PD by a neurologist and consulted to be submitted to the rehabilitation part for VFSS due to subjective dysphagia. Patients who possessed facial muscle strength sufficient to bite a cylindrical mouthpiece and the cognition to exhale with maximum strength were enrolled in the study. Patients with a history of stroke or other neuromuscular diseases that could affect other respiratory muscles, chronic obstructive pulmonary disease, asthma, other chronic respiratory diseases, neck tumors, or head tumors, were excluded from the study. Following exclusions, this study possessed a total of 219 eligible patients.

All participants provided written informed consent to participate in this study. The Institutional Review Board of Dong-A University Hospital reviewed and approved this study (DAUHIRB-EXP-23-035, approval date 22 February 2023).

### 2.2. Methods

VFSS is a standard diagnostic tool for identifying dysphagia and aspiration. The VFSS includes visualization of the oral cavity, pharynx, larynx, upper esophageal structures, and physiologic contents of the process during swallowing. In our study, VFSS protocol was performed based on the modified Logemann’s protocol [25], and the examination was performed in the fluoroscopic X-ray room of our clinical center by an experienced doctor. The patient maintained a sitting position, obtained a lateral view, and tested with barium contrast (barium sulfate; Solotop suspension) mixed with foods of various viscosities from liquid to solid. The patient was asked to swallow curd-type yogurt 2 times, rice porridge 3 times, then boiled rice 3 times. Then asked to swallow thin liquid using spoon, straw, and cup 3 times each. All foods were mixed with undiluted barium, and thin liquid was 35% diluted barium solution. Any abnormalities were recorded, and the test was stopped when severe aspiration was observed. At the same time, a video image was recorded at a speed of 30 frames per second, and after the VFSS test, the recorded video was analyzed through the consent of several experienced rehabilitation doctors. Based on the VFSS image, the FDS score and the PAS score were set.

PAS is an 8-point ordinal scale that confirms the presence and depth of penetration and aspiration. Higher PAS scores indicate a more severe swallowing dysfunction. PAS 1 was defined as normal, PAS 2–5 denotes laryngeal penetration, and PAS 6–8 denotes airway aspiration (Table 1) [26]. Based on the VFSS test, the PAS score was recorded as the highest score during the test.

PCF was tested at the start of the study, using a PCF meter (PF100; Microlife Corp., New York, NY, USA) equipped with a cylindrical disposable mouthpiece to obtain measurements. Following training on the measurement method, three trials were performed, with the average value of the PCF meter recorded. Subsequently, VFSS was performed to obtain the PAS scores. Classification of the patients was enacted based on PAS scores, with patients who had aspiration in VFSS classified into the aspiration group (PAS ≥ 6), and patients who did not have aspiration in VFSS classified into the non-aspiration group (PAS ≤ 5).

### 2.3. Statistical Analysis

IBM SPSS Statistics (version 22.0; IBM SPSS, Armonk, NY, USA) and MedCalc (version 20.015; MedCalc Software, Ostend, Belgium) were utilized for statistical analyses. A *t*-test was used to compare the basic characteristics between the two groups. The cut-off value was obtained using the Youden index through the receiver operating characteristic (ROC) curve of PCF. Additionally, univariate and multivariate analyses of the logistic regression test were used to determine aspiration risk. For the *t*-tests, univariate analyses, and multivariate analyses, statistical significance was defined as a *p*-value of less than 0.05.

## 3. Results

Of the 219 patients enrolled in the study, 125 patients were classified into the aspiration group, and 94 patients were classified into the non-aspiration group. The baseline characteristics of the two groups’ age, sex, body mass index (BMI), Hoehn and Yahr scale, mini-mental state examination (MMSE), smoking history, PCF, history of hypertension, and diabetes, were compared. Between the characteristics of the two groups, there were significant differences in sex, BMI, and the Hoehn and Yahr scale. Statistical significance was also observed in PCF. Age and MMSE were observed similarly in two groups (Table 2).

From the mean PCF value comparison, a significantly lower value was observed in the aspiration group (non-aspiration group, 181.38 ± 103.92; aspiration group, 132.63 ± 83.62; *p* < 0.001) (Figure 1). Additionally, the ROC curve analysis revealed a cut-off value of 153 L/min using the Youden index. The area under the curve (AUC) value was 0.648, the *p*-value was less than 0.001 (Figure 2), the sensitivity was 73.06%, and the specificity was 51.06% (Table 3).

A univariate analysis was performed with baseline characteristics to determine whether a PCF of 153 or less could act as a risk factor for aspiration, and statistically significant results were found for the male sex, BMI, Hoehn and Yahr scale scores, and PCFs of 153 or less. To a similar degree, in the multivariate analysis, significant results were observed in the male sex, Hoehn and Yahr scale scores, and PCFs of 153 or less. The comparative risk for PCFs of 153 or less had an odds ratio (OR) of 3.648, indicating a higher comparative risk compared to other variables (Table 4).

## 4. Discussion

This study investigated the relationship between aspiration and PCF in patients with PD. Previous studies have reported a relationship between cough and aspiration risk in PD [24,27,28,29]. For instance, Silverman et al. [14] examined the correlation between voluntary PCF, dysphagia severity, and PD severity. Higher PAS scores were found to be related to lower voluntary PCF and positively correlated with disease severity [14]. However, a fundamental limitation was that the cut-off value of the PCF for the risk of aspiration was not reported. Curtis et al. [12] previously suggested a cut-off value of 42.5 L/min for the risk of airway invasion in PD. However, this value was measured through the cough reflex using capsaicin, which has a key limitation of causing patient discomfort. Additionally, the criterion for the PAS score was 3, which is not very useful in clinical practice, and the study had a small sample size of only 22 participants. This study used a PAS of 6 or higher as the criterion, elevating the association with pneumonia. Compared to previous studies, a large population size of 219 patients was included, increasing the clinical significance. Furthermore, a significant correlation was noted between voluntary PCF and the risk of aspiration, and a cut-off value of 153 was observed. These findings suggest that our study is clinically useful and effective.

In a previous study, the clinical risk factor of dysphagia in patients with PD were a score greater than 3 on the Hoehn and Yahr scale, related weight loss or a BMI less than 20, drooling or hypersecretion, and dementia [30]. The present study evaluated the relative risk through a logistic regression analysis by comparing the Hoehn and Yahr scale, BMI, and MMSE with a PCF of 153 or less. A multivariate analysis revealed that the male sex, Hoehn and Yahr scale scores, and PCF of 153 or less were significantly correlated with aspiration (Hoehn and Yahr Scale OR; 1.959, *p* < 0.001, PCF ≤ 153; OR 3.648, *p*< 0.001). BMI showed an OR of 0.912 and was significant in a univariate analysis but not in a multivariate analysis. Moreover, the MMSE score did not show statistical significance in a univariate analysis. In this study, a PCF below the cut-off value of 153 had a higher odds ratio than the previously known risk factor for dysphagia, and was also statistically significantly correlated for aspiration in PD.

Respiratory muscle weakness accompanies PD progression [13,14]. The respiratory and swallowing muscles are innervated by similar areas, and previous studies have demonstrated that PCF in PD is related to maximum inspiratory pressure and disease severity [4,14]. Furthermore, one study showed that expiratory muscle strengthening training improves swallowing function and spontaneous coughing [13]. Therefore, it is important to indirectly evaluate the swallowing muscles through the expiratory muscles and indirectly measure disease severity [14].

The PCF cut-off value of 153 derived in this study was similar to the standard value of 160 for decannulation after tracheotomy [31]. Previous studies on patients with strokes have also revealed that dysphagia is associated with PCF, and a study by Han et al. suggested a cut-off value of 151 [32,33]. The cut-off value of 153 in this study was similar to the PCF reference values of previous studies, suggesting clinical significance.

This study measured PCF using a simple handheld machine, and statistical significance was demonstrated with appropriate statistical tests. When evaluating dysphagia in patients clinically, it is common to conduct bedside swallowing evaluations first and accordingly select patients for VFSS. PCF can be actively used in such bedside assessments. Currently, not many bedside swallowing function assessment tools can help to measure coughing. To date, only one clinical tool, the Mann Assessment of Swallowing Ability (MASA), has been reported for subjectively evaluating the cough reflex [14,34]. Further studies are required to determine whether assessing PCF can positively influence the sensitivity and validity of existing assessment tools.

In a multivariate analysis, the male sex was found to be statistically significant. In previous studies, the correlation between the male sex and aspiration in patients with PD was not prominent. However, several previous studies reported that the male sex was more related to dysphagia in PD. Cereda et al. reported that older age, the male sex, a longer disease duration and the presence of dementia were clinical predictors of dysphagia in PD patients [35]. Additionally, Nienstedt et al. identified a higher age, an affirmed subjective aspiration sign, and the male sex as predictors of dysphagia in PD. In a study by Claus et al., researchers tried to identify clinical predictors indicating pharyngeal dysphagia in PD patients using a flexible endoscopic evaluation of swallowing. They also found that the male sex was more related to dysphagia than the female sex [36]. Physiological striatal dopamine levels is higher in females, due to estrogen activity [37,38]. This kind of physiological differences, according to sex in PD, might affect dysphagia, so the male sex may be a risk factor for PD-related dysphagia. This study had a few key limitations. First, this was a retrospective study, so the quality of the collected data might have affected the reliability of the results. Furthermore, it could be difficult to adjust all related factors which could influence the results. Second, selection bias could have occurred when choosing the patients. Patients with severe or mild PD may not be included in the study. However, if the purpose is to screen for dysphagia using PCF values, the bias of data collection is considered to be insignificant. Third, the reliability of PCF may have been influenced by the patients’ alertness or cognition at the time of testing. Fourth, our PCF machine is unable to record values under 50, and these were all recorded as 0. Fifth, this retrospective study can only identify the association between clinical factors and lacks a cause-and-effect relationship. However, the temporal–precedence relationship is not clear, and the results of this study are considered to have clinical significance in using it as a tool for screening, whether a diagnostic test such as VFSS is necessary. Through this proposed retrospective pilot study, we were able to find out the association between PCF and aspiration in PD patients. Future prospective studies with a larger sample size are needed to investigate the association between PCF and dysphagia with more reliability and accuracy.

## 5. Conclusions

This study found that in patients with PD, more severe dysphagia is associated with lower voluntary PCF. Furthermore, a PCF of 153 or less was confirmed to be a risk factor for aspiration. Therefore, when evaluating dysphagia in patients with PD, PCF can be used to indirectly screen for the presence and severity of dysphagia and can be used as an indicator for further comprehensive evaluation.

## Figures and Tables

**Figure 1 medicina-59-00921-f001:**
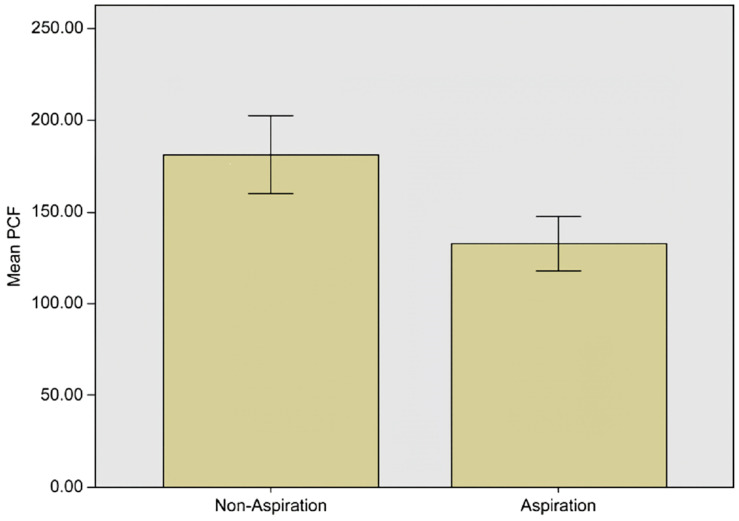
PCF values of the non-aspiration group and the aspiration group.

**Figure 2 medicina-59-00921-f002:**
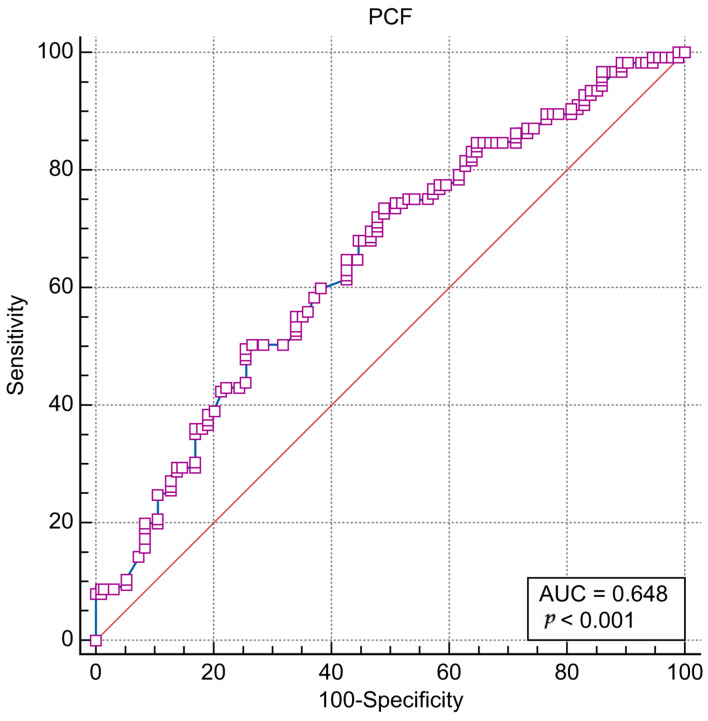
Receiver operating characteristic (ROC) curve analysis for the cut-off values of PCF (L/min).

**Table 1 medicina-59-00921-t001:** Penetration–Aspiration Scale (PAS).

Score	Description of Events
1	Material does not enter the airway.
2	Material enters the airway, remains above the vocal folds, and is ejected from the airway.
3	Material enters the airway, remains above the vocal folds, and is not ejected from the airway.
4	Material enters the airway, contacts the vocal folds, and is ejected from the airway.
5	Material enters the airway, contacts the vocal folds, and is not ejected from the airway.
6	Material enters the airway, passes below the vocal folds, and is ejected into the larynx or out of the airway.
7	Material enters the airway, passes below the vocal folds, and is not ejected from the trachea despite effort.
8	Material enters the airway, passes below the vocal folds, and no effort is made to eject.

**Table 2 medicina-59-00921-t002:** Characteristics of study participants between aspiration group and non-aspiration group.

	Aspiration Group (*n* = 125)	Non-Aspiration Group (*n* = 94)	*p*-Value
Age (years)	73.99 ± 8.87	71.88 ± 9.54	0.093
Sex (Male/Female)	(81/44)	(46/48)	0.019 *
BMI (kg/m^2^)	21.48 ± 3.79	22.78 ± 3.72	0.012 *
Hoehn and Yahr scale	3.45 ± 0.85	2.82 ± 0.92	<0.001 *
MMSE	21.18 ± 6.17	22.76 ± 6.83	0.075
Smoking	32 (25.6%)	17 (18.1%)	0.181
PCF	132.63 ± 83.62	181.38 ± 103.92	<0.001 *
Hypertension (HTN)	41 (32.8%)	36 (38.3%)	0.401
Diabetes mellitus (DM)	24 (19.2%)	22 (23.4%)	0.452

BMI: body mass index; MMSE: mini-mental state examination; PCF: peak cough flow; HTN: hypertension; DM: diabetes mellitus. Values are presented as numbers (%) or mean ± standard deviation. * *p* < 0.05.

**Table 3 medicina-59-00921-t003:** Diagnostic parameters of the cut-off value of PCF.

PCF	Asp (+)	Asp (−)	Sensitivity	Specificity	PPV	NPV
PCF ≤ 153	92	46	73.60 (65.0–81.1)	51.06 (40.5–61.5)	66.7 (61.3–71.6)	59.3 (50.5–67.4)
PCF > 153	33	48				

Cut-off values determined by ROC curve analysis with the Youden index. Results are presented as values with 95% confidence intervals. PCF, peak cough flow; PPV, positive predictive value; NPV, negative predictive value.

**Table 4 medicina-59-00921-t004:** Logistic regression of aspiration risk in Parkinson’s disease (*n* = 125).

Variable	Univariate Analysis		Multivariate Analysis	
	OR (95% CI)	*p*	OR (95% CI)	*p*
Age (per 1 year)	1.025 (0.996–1.056)	0.095	N/A	
Male sex	1.921 (1.113–3.317)	0.019 *	3.264 (1.630–6.535)	0.001 *
BMI (per 1 kg/m^2^)	0.912 (0.848–0.982)	0.014 *	0.931 (0.857–1.012)	0.095
Hoehn and Yahr scale	2.232 (1.598–3.117)	<0.001 *	1.959 (1.379–2.783)	<0.001 *
MMSE	0.961 (0.920–1.004)	0.077	N/A	
Smoking	1.559 (0.805–3.019)	0.188	N/A	
PCF ≤ 153	2.909 (1.650–5.130)	<0.001 *	3.648 (1.797–7.407)	<0.001 *
Hypertension	0.786 (0.450–1.375)	0.399	N/A	
Diabetes mellitus	0.778 (0.405–1.494)	0.450	N/A	

OR, odds ratio; CI, confidence interval; BMI, body mass index; MMSE, mini-mental state examination; PCF, peak cough flow. * *p* < 0.05.

## Data Availability

Not applicable.

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
