# Peer review of "Cut-Off Value of Voluntary Peak Cough Flow in Patients with Parkinson’s Disease and Its Association with Severe Dysphagia: A Retrospective Pilot Study"

_medicina, 2023, doi:10.3390/medicina59050921_

Round 1

Reviewer 1 Report

For The review of dysphagia and PD with correlation to male , lower body mass index and hoehn and yahr scores cannot be inferred based on a retrospective analysis that does not have an actual modified barium swallow study documented with level of penetration of food and documented objective aspiration.

Consider changing the paper title to a proposed pilot

study based on retrospective review.

Also the study is not very clear about the methodology: where did you collect the data , why choose a retrospective study

The methodology is very unclear: how did you conclude aspiration risk wothout radiology results and why just use PAS

why male gender amd lower body weight- lower weight was a predictor of dyspahgia because they were already having difficulties swallowing and thus had lower weight

This can be regitled as a proposal based on limited data

acknowledge the following limitations and provide more rationale

explain limitations of a retrospective review.

A retrospective review means that this is not a study but review of previous data without a null hypothesis.

consider renaming this a review article and a proposed study.

Author Response

Response to Reviewer 1 Comments

Point 1: For The review of dysphagia and PD with correlation to male , lower body mass index and hoehn and yahr scores cannot be inferred based on a retrospective analysis that does not have an actual modified barium swallow study documented with level of penetration of food and documented objective aspiration.

Response 1:

In my manuscript, I felt that the detailed explanation of VFSS was lacking, so I added details and protocols about the test. VFSS is a standard diagnostic tool that can objectively evaluate dysphagia with fluoroscopy using barium, and radiologic and objective evaluation of aspiration is possible. Through VFSS, the degree of penetration and aspiration was evaluated with the PAS score.

Point 2:

Consider changing the paper title to a proposed pilot

study based on retrospective review.

Response 2: Thanks to reviewer’s opinion, we changed our title like this.

Cut-Off Value of Voluntary Peak Cough Flow in Patients with Parkinson’s disease and its Association with Severe Dysphagia: A Retrospective Pilot Study

Point 3: Also the study is not very clear about the methodology: where did you collect the data, why choose a retrospective study

Response 3: In methodology, the contents were supplemented and explained that we reviewed the medical records of our tertiary hospital rehabilitation center.

Enrolled patients were diagnosed with PD by a neurologist and consulted to rehabilitation part for VFSS due to subjective dysphagia And our study was conducted as a retrospective pilot study for a future prospective larger study.

Point 4: The methodology is very unclear: how did you conclude aspiration risk wothout radiology results and why just use PAS

Response 4: It is considered that the answer to the point 1 can also be answered in this point.

Point 5 : Why male gender amd lower body weight- lower weight was a predictor of dyspahgia because they were already having difficulties swallowing and thus had lower weight

Response 5: We agree that there is a logical leap that male sex and low BMI are predictors of dysphagia. The cause effect relationship cannot be revealed in this study. However, we tried to reveal the correlation of dysphagia and other clinical variables by multivariate analysis through retrospective data in this study.

So we changed the words "predictor" to "risk factor" or "correlation" in our manuscript, which dosen't contain the cause effect relationship or temporal relationship.

The purpose of this study is to identify which clinical factor can be used as screening tool for aspiration and when to do definite diagnostic test like VFSS, so it is considered to have clinical meaning.

Point 6: This can be regitled as a proposal based on limited data

acknowledge the following limitations and provide more rationale

explain limitations of a retrospective review.

 Response 6: In the manuscript, we supplemented the limitations of the retrospective study.

The study was based on the limited data of the retrospective study, so the reliability of the data may be low, and selection bias may occur, and also that the cause-effect relationship is not clear.

Point 7: A retrospective review means that this is not a study but review of previous data without a null hypothesis.

Response 7 : While partially agreeing with the reviewer's opinion, in fact, we conducted a study, not a retrospective review, with the hypothesis that low PCF is related to severe dysphagia (aspiration) in PD patients, and that a cut-off value of PCF could be revealed through this study.

Our retrospective study will verify the hypothesis and confirm through a prospective study in the future. We have added details to further clarify our hypotheses.

Point 8: consider renaming this a review article and a proposed study.

Response 8: As we answered at point 2, we changed our title.

And also we supplemented our manuscript to reveal that this is proposed pilot study.

Reviewer 2 Report

In this paper, the authors address a relevant clinical issue in Parkinson's patients. Impairment of buccofacial function leads to malnutrition, weight loss, and difficulties with medication and puts patients at risk of pulmonary infection by aspiration. The main finding of this paper is that a low peak cough flow is a risk factor for aspiration in patients with Parkinson's disease. Furthermore, a cut-off value is defined for low peak cough flow. 

The study is performed in a monocentric retrospective design. The number and selection of participants are adequate. The applied statistical analysis is suited to conclude the addressed question. Limitations are addressed, and findings are put into relation. 

The method used to measure peak flow by a handheld instrument can easily be applied in the clinical context, and defined cut-off volume gives an additional instrument to help change, for example, the medical regime. With the large cohort, a more reliable cut-off value for this clinical usage is defined. An imbalance in the compared groups with a higher proportion of males in the aspirational group could be the reason for identifying sex as an influencing factor. Authors should put in a sentence to clarify this in the discussion.  

The work is a valuable contribution to the basic clinical care for Parkinson's patients. I would recommend accepting the paper for publication. 

Minor corrections should be made, for example, inconsistency of hyphenation (cut-off vs cut-off) and capitalisation (Receiver vs receiver).

Author Response

Response to Reviewer 2 Comments

Point 1: An imbalance in the compared groups with a higher proportion of males in the aspirational group could be the reason for identifying sex as an influencing factor. Authors should put in a sentence to clarify this in the discussion.  

Response 1: Thank you for your comments and I sincerely agree with reviewer’s opinion.

The characteristics observed in table 2 are the result of classified group, according to presence of aspiration after VFSS. I wrote as “baseline characteristics” in my manuscript, and I think it might cause some misunderstandings. So I deleted the word “baseline” and modified some sentences.

Actually we researched more previous studies about correlation of male sex and dysphagia, and we additionally found some related studies. Several studies found that male sex is more related in Parkinson’s disease, because physiological striatal dopamine levels is higher in female, due to oestrogen activity. These studies reported male sex was significantly associated with dysphagia in Parkinson. So I supplemented theses contents in my manuscript. I think it could be the answer of why male sex was found to be statistically significant in multivariate analysis.

Round 2

Reviewer 1 Report

The revisions are adequate.